# Neutrophil–Lymphocyte Ratio Values in Schizophrenia: A Comparison between Oral and Long-Acting Antipsychotic Therapies

**DOI:** 10.3390/brainsci14060602

**Published:** 2024-06-14

**Authors:** Antonino Messina, Fabrizio Bella, Giuliana Maccarone, Alessandro Rodolico, Maria Salvina Signorelli

**Affiliations:** 1Psychiatry Unit, Department of Clinical and Experimental Medicine, University of Catania, 95124 Catania, Italy; thomessina@gmail.com (A.M.); giuliana.mcc@gmail.com (G.M.); alessandro.rodolico@me.com (A.R.); maria.signorelli@unict.it (M.S.S.); 2Psychiatry Unit, Department of Mental Health, ASP Enna, 94100 Enna, Italy

**Keywords:** schizophrenia, neuroinflammation, biomarkers, long-acting antipsychotics

## Abstract

Background: Schizophrenia is a mental disorder affecting approximately 0.32% of the global population, according to the World Health Organization. Antipsychotic medications are used to treat this condition by inhibiting D2 dopamine and 5HT2 serotonin receptors. The selection of the appropriate mode of delivery for these drugs is based on factors such as patient adherence, clinical presentation, and patient preferences. However, additional drivers of treatment selection are required in clinical practice. Mounting evidence suggests that neuroinflammation plays a crucial role in the pathogenesis of schizophrenia. NLR, a cost-effective biomarker of inflammation, has increased in several psychiatric conditions and may represent a valid method for studying the inflammatory stage in schizophrenia, relapse, and the first episode of psychosis. The aim of this study is to evaluate whether there are any variations in NLR values between patients given oral antipsychotics and those given long-acting antipsychotics. Methods: The study included 50 individuals with schizophrenia, either acute or in the follow-up phase. NLR was obtained by calculating the ratio of absolute neutrophil count (cells/μL) and absolute lymphocyte count (cells/μL). Results: Patients on long-acting antipsychotics exhibited significantly lower mean NLR scores (1.5 ± 0.7) compared to those on oral antipsychotics (2.2 ± 1.3) (*p* < 0.05). Conclusions: NLR appears promising as a neuroinflammatory biomarker. This study reveals significantly lower NLR values in patients on long-acting antipsychotics, which may signify reduced systemic inflammation and improved adherence.

## 1. Introduction

Schizophrenia is a chronic and severe mental disorder that affects how a person thinks, feels, and behaves. Signs and symptoms can differ, but they typically include delusions, hallucinations, severely disorganized thinking and behavior, or negative symptoms, all of which indicate a reduced ability to function. Prodromal symptoms often appear before the active phase, and residual symptoms may follow, featuring mild or less intense hallucinations or delusions [1]. According to the World Health Organization, about 24 million people worldwide, or 0.32 percent of the world’s population, suffer from schizophrenia [2]. Medical treatment of this condition comprises the use of antipsychotic medications [3]. Most of these drugs inhibit D2 dopamine receptors, therefore decreasing dopaminergic neurotransmission to reduce positive symptoms of schizophrenia, and inhibit 5HT2 serotonin receptors, thus reducing negative symptoms; other molecules, such as aripiprazole, cariprazine, and brexpiprazole, possess different properties regarding receptor profiles, acting as partial agonists at D2 receptors [4]. Aripiprazole also possesses the peculiarity of acting as a prevalent antagonist at D2 receptors when dopamine is highly concentrated in the synapse, while acting as a partial agonist when the synaptic concentration of dopamine is scarce; also, aripiprazole has been shown to behave as a partial agonist at 5HT1A receptors and as an antagonist at 5HT2A receptors [5]. Antipsychotics are classified into two groups, i.e., first-generation antipsychotics (FGA) and second-generation antipsychotics (SGA). FGA includes medications such as haloperidol and chlorpromazine. They are mainly effective in alleviating positive symptoms (hallucinations and delusions) but are more likely to cause extrapyramidal side effects (motor control issues like tardive dyskinesia). SGA includes medications such as risperidone, olanzapine, quetiapine, and aripiprazole. They are effective in treating both positive and negative symptoms and generally have a lower risk of extrapyramidal side effects. However, they may increase the risk of metabolic syndrome (weight gain, diabetes, hyperlipidemia). Antipsychotics can be administered in three alternative formulations, i.e., oral, sublingual, and injectable or long-acting. Oral and injectable forms are used to treat psychotic disorders like schizophrenia, but they differ significantly in their administration, pharmacokinetics, adherence implications, and clinical applications. Oral antipsychotics have greater flexibility in managing side effects by adjusting the dose quickly, but, as they must be taken daily, treatment adherence needs to be considered. Long-acting injectable (LAI) formulations, such as risperidone, paliperidone, aripiprazole, and olanzapine, are administered via intramuscular injection at intervals ranging from every 2 weeks to every 3 months, depending on the specific medication. While they offer less flexibility in dose adjustments compared to oral medications, they improve adherence due to less frequent dosing, thereby reducing the risk of relapse associated with missed doses [6]. Sublingual antipsychotics, such as asenapine, offer an alternative route of administration for patients who have difficulty swallowing pills or experience gastrointestinal issues with oral medications. They are not as widely available as oral or injectable antipsychotics and may have fewer options in terms of medication choices. Criteria such as patient’s adherence, clinical presentation, and patient’s preferences are used to establish the most appropriate mode of delivery [7]. However, further drivers of treatment selection are required in clinical practice [8,9].

Mounting evidence highlights the role of microglia and neuroinflammation as the pathogenic basis of schizophrenia [10,11]. Indeed, a plethora of stimuli results in the activation of the immune system with the production of cytokines, ultimately leading to microglial activation and establishing a neuroinflammatory process that can progressively evolve into neurodegeneration. Some brain areas are more susceptible to neuroinflammation than others, and their dysfunction underlies the neuroanatomical basis of schizophrenia [12,13]. Specifically, it has been demonstrated how elevated levels of inflammation (considering markers such as interleukin-6, C-reactive protein) correlate with decreased activity in limbic regions (i.e., amygdala, hippocampus, anterior insula, temporal pole) and more significant connectivity between the hippocampus and the medial prefrontal cortex. Consequently, assessment of neuroinflammatory status can have a major role in the screening and monitoring of schizophrenia [10]. Among different markers of neuroinflammation, interleukin-6 (IL-6), tumor necrosis factor (TNF-alpha), C-reactive protein (CRP), and neutrophil/lymphocyte ratio (NLR) are the most investigated [14].

NLR is a cost-effective biomarker of inflammation that is increased in several psychiatric conditions [15,16]. Specifically, the association between NLR and schizophrenia has been shown [17], and several data suggest that NLR can represent a valid method for studying the inflammatory stage in schizophrenia, in relapse, and in the first episode of psychosis [18]. High NLR levels are independent of metabolic parameters, suggesting that inflammatory processes might contribute to the disease’s etiology [19]. Moreover, NRL could play a crucial role in the identification of forms of schizophrenia with greater neuroinflammatory components and therefore guide treatment selection.

Long-acting antipsychotics represent an effective alternative to oral therapy, which has shown to determine clinical advantages for what concerns clinical response, re-hospitalization, mortality, and overall outcome [20,21]. It is well known from recent literature that antipsychotics, especially long-acting formulations, possess immunomodulatory activity that is expressed through remodulation of pro-inflammatory pathways [22,23,24].

This pilot study seeks to determine whether there are any variations in NLR values between patients given oral antipsychotics and those given long-acting antipsychotics. Given the anti-inflammatory effects of some antipsychotics, such as paliperidone and aripiprazole, we expect that the greater stability in plasma levels provided by LAIs might result in lower inflammatory levels compared to oral administration.

## 2. Materials and Methods

### 2.1. Study Design and Participants

A retrospective study, based on chart review, was carried out at the Department of Clinical and Experimental Medicine, Institute of Psychiatry, University of Catania (Catania, Italy). All patients signed informed consent to the utilization of clinical data from clinical records for research purposes.

The study included 50 individuals with schizophrenia, either acute or in partial or full remission, who referred to our Institute in the last six months, i.e., with a recent history of illness. Schizophrenia was diagnosed according to the DSM-5-TR criteria. Only patients using a long-acting antipsychotic (aripiprazole or paliperidone) on a monthly administration and patients taking only oral aripiprazole or paliperidone were included. Among the other antipsychotic drugs, paliperidone and aripiprazole were chosen to conduct this study due to their good tolerance profiles as well as their known and consistent anti-inflammatory effects [23,25]. Specifically, paliperidone administration is related to decreased levels of IL-1B, reduction of inflammasome complexes, and down-regulation of NF-KB and iNOS pathways while up-regulating endogenous anti-inflammatory and antioxidant bio-molecular pathways [26,27,28]. Analogously, aripiprazole exerts anti-inflammatory activity, which is observable through the reduction of IL-1B, TNF-a, IL-6, TGF-b1, and other pro-inflammatory mediators [29]. It has also been shown that aripiprazole administration directly determines decreased mRNA levels of IL-1B, IL-6, and TNF-a while up-regulating mRNA levels of IL-10, which is known to exert anti-inflammatory effects [30]. Moreover, a recent review that compared the effect of antipsychotics on the immune system pointed out that aripiprazole exerts a greater reduction in pro-inflammatory interleukins, PGE2, COX2, and NO compared to other antipsychotic drugs; in the same review, authors highlighted how paliperidone, in addition to the aforementioned reduction in pro-inflammatory markers, exerts an increase in BDNF levels [23].

Other inclusion criteria were as follows: age 18–65 years; males or females during the mid-luteal phase of the menstrual cycle, which has been reported to be immunologically neutral [31]. The mid-luteal phase was assessed by averaging the duration of each woman’s three luteal phases. This was calculated by identifying, in the last three menstrual cycles, the date of onset of menstruation, the ovulatory phase at 14 days, and the date of onset of the next menstrual cycle. The period between ovulation and the start date of the next cycle corresponds to the luteal phase. Averaging the duration of the last three luteal phases allowed us to identify the mid-luteal phase. Exclusion criteria were as follows: major clinical conditions; presence of infective diseases in the last two months; concomitant drugs; substance abuse; women in the follicular phase (menstruation or ovulation).

### 2.2. Data and Sample Collection

Clinical data were collected from patients’ charts. Participants were recruited during first or follow-up psychiatric visits conducted at our outpatient services. After verifying eligibility based on the aforementioned inclusion and exclusion criteria, the study’s procedures and objectives were explained to the patients, asking for their willingness to participate. Upon signing the informed consent, a second appointment was scheduled to collect blood samples, performed at 7 a.m. as a routine examination. NLR was obtained by calculating the ratio of absolute neutrophil count (cells/μL) and absolute lymphocyte count (cells/μL). The normal range of NLR is unanimously referred to be between 1–2, indicating an equilibrium in immunological state. Values higher than 3.0 and below 0.7 are pathological in adults [32].

### 2.3. Statistical Analysis

Data were reported by mean and standard deviation for continuous variables, and categorical variables were reported as percentages. Differences between patients on oral antipsychotics and those on long-acting antipsychotics were investigated with a pure exploratory intent using the chi-square test or the Student’s *t*-test, as appropriate. To further investigate the differences in neutrophil-to-lymphocyte ratio (NLR) between the two groups, a multivariate regression analysis was performed. This analysis adjusted for potential confounding demographic variables, including age, gender, and duration of illness. The regression model included NLR as the dependent variable and treatment type (long-acting vs. oral) as the primary independent variable, along with the demographic covariates. A *p* value <0.05 was considered statistically significant. Statistical analyses were carried out using SPSS statistical software, version 28 (IBM Corp., Released 2021, IBM SPSS Statistics for Windows, Version 28.0. IBM Corp., Armonk, NY, USA).

## 3. Results

Twenty-seven patients were taking second-generation oral antipsychotics (aripiprazole 10 mg or paliperidone 6–9 mg), and twenty-three of them took long-acting antipsychotics (aripiprazole 400 mg and paliperidone 75–150 mg). Patients included in our study were taking antipsychotic therapy for at least one year. The average duration of illness in patients on oral therapy was 3 years, while for patients undergoing long-acting therapy it was 2 years (see Table 1). None of the patients enrolled in our study suffered from major medical conditions and/or infective diseases that could have affected NLR.

Patients were homogeneous in the two groups in terms of gender, age, and marital status (see Table 1). The duration of psychosis for patients on long-acting antipsychotics was significantly shorter compared to those on oral antipsychotics (*p* = 0.03). The mean NLR score for patients on long-acting antipsychotics was significantly lower than that for patients on oral antipsychotics (*p* = 0.03). Regarding gender, there was no significant difference between the two groups (*p* = 0.08). Additionally, no differences were found between individuals receiving oral/long-acting aripiprazole and those receiving oral/long-acting paliperidone (*p* = 0.83).

The ANCOVA results (Table 2) showed that the effect of treatment type on NLR remained statistically significant (*p* = 0.008) even after controlling for age, gender, duration of psychosis, PANSS, marital status, and drug type. Other variables such as age (*p* = 0.688), duration of psychosis (*p* = 0.219), PANSS (*p* = 0.243), gender (*p* = 0.569), marital status (married: *p* = 0.241, single: *p* = 0.521), and drug type (paliperidone: *p* = 0.860) did not significantly affect NLR.

## 4. Discussion

To our knowledge, this is the first study comparing NLR values between patients taking oral antipsychotics and patients treated with long-acting antipsychotics. NLR represents a reliable biomarker of immune activation and could be used in monitoring those forms of schizophrenia in which neuroinflammation is present. The production of pro-inflammatory cytokines, which are secreted by neutrophils, alters the permeability of the blood-brain barrier, resulting in hyperactivation of microglia, subsequent neuroinflammation, and neurodegeneration, characterized by neuronal death and reduction of the dendritic tree. The effect of antipsychotics could control and reduce neuroinflammatory processes and, mainly if administered at an early stage of psychosis, could prevent the damage caused by the activation of neuroinflammatory pathways from being overlaid by neurodegenerative processes, with progressive worsening of the clinical pattern and a poorer prognosis.

Inconstant NLR values between the two population groups could be due to several factors. In normal hemostatic conditions, only a few neutrophils are found in organs such as the liver and adipose tissues [33]. In a large cross-sectional retrospective study, authors reported that after treatment with antipsychotics, NLR is reduced [34,35]. Various possible hypotheses exist for reduced NLR readings in patients treated with long-acting antipsychotics. Lower readings, for example, could be attributable to decreased systemic inflammation, and it is known from literature that long-acting antipsychotics have a stronger anti-inflammatory impact than oral antipsychotics [36]. Furthermore, long-acting antipsychotics are associated with sustained and constant medication release, resulting in stable plasma levels and a more extended anti-inflammatory action [37]. This idea is supported by some authors [38], who found that particularly aripiprazole and paliperidone have consistent anti-inflammatory effects. These drugs reduce systemic inflammation, thereby decreasing NRL readings [39]. In a neuroinflammation model, paliperidone administration has demonstrated anti-inflammatory and antioxidant activity [26,40]. Improved medication adherence is another explanation for reduced NLR values in patients treated with long-acting antipsychotics: improved medication adherence results in a better control of psychotic symptoms and a considerable reduction in the inflammatory response in individuals using long-acting antipsychotics [41]. On the other hand, poor medication adherence associated with the use of oral antipsychotics may result in uncontrolled symptoms, an increase in the inflammatory response, and higher NRL levels [42].

On the other hand, another mechanism by which antipsychotics administration could result in modifications of NLR is through drug-induced neutropenia. This phenomenon, although most often associated with clozapine intake, could occur following the intake of other atypical antipsychotics, including paliperidone, for which a reduction in neutrophil and lymphocyte counts was observed [23,43]. Moreover, even fluctuations in lymphocyte blood levels could be directly due to disease state, independently of antipsychotic medications [44].

Moreover, discrepancies in NLR values between the two groups may be related to patient population differences [45]. Patients treated with oral antipsychotics, for example, may have had more severe or treatment-resistant symptoms than those treated with long-acting antipsychotics. Because of their underlying illness, these patients may have a more severe inflammatory reaction. Neuropathological signs of chronic neuroinflammation and associated neurodegeneration are significantly related to higher levels of NLR. Also, psychotic patients who relapse more frequently have higher NLR values [46]. The use of oral therapy, as effective as it is, requires daily administration, which in poorly compliant patients becomes critical.

We are aware of the possibility that patients with different severities might have divergent NLR values [47]. For this reason, we checked if the PANSS score had an impact on the difference in NLR between patient groups and found no difference.

## 5. Conclusions

NLR values were significantly lower in blood samples from patients with long-acting antipsychotic treatment compared to those treated with oral antipsychotics, according to this study. This outcome could be attributed to decreased systemic inflammation and improved medication adherence.

### Limitations

Obviously, the present study has some limitations, including the retrospective design of the study, the small sample size, the lack of a healthy control group, and the exclusion of deeper bio-molecular analyses, which may clarify whether lowering of NLR represents an epiphenomenon of LAI administration or whether there is an actual modulation on the brain and peripheral immune system. In fact, it is known that administration of antipsychotics may induce demarginalization of neutrophils, which may consequentially impact the NLR. Further research should be conducted to confirm these findings and investigate in a larger case-control study the activity of oral and long-acting antipsychotics on NLR; also, it would be possible to investigate the rationale for early use of long-acting antipsychotics at the onset of schizophrenia, studying neuroinflammatory patterning at diagnosis and during monitoring to assess the effectiveness of therapy.

Another limitation may be due to the choice of the two molecules paliperidone and aripiprazole: in fact, as already discussed, aripiprazole exerts partial agonism at D2 receptors, while paliperidone is a pure antagonist. Further studies should consider antipsychotics with similar mechanisms of action (e.g., aripiprazole vs. brexpiprazole) to relieve this possible confounding factor.

Also, a larger study would consent to accumulate data that can clarify whether the NLR can be considered a marker associated with certain well-defined psychopathological domains and whether it can be considered a prognostic factor for cognitiveness and global functioning. These preliminary data could form the basis for clinical implications, as the NLR could be employed to monitor the progress of schizophrenia, potentially distinguishing between acute and status phases. Lastly, further studies in this direction, with greater sample sizes and bio-molecular analyses, could allow us to identify new pharmacological tools that target the molecular pathways related to neuroinflammation, with the possibility to intervene in the early stages of disease by preventing the establishment of a chronic inflammatory state.

## Figures and Tables

**Table 1 brainsci-14-00602-t001:** Patient characteristics and variables.

	Oral Antipsychotics (n = 27)	Long-Acting Antipsychotics (n = 23)	*p*-Value
Males, number (%)	11 (40.7)	16 (69.6)	0.08
Mean age, years (SD)	41.3 (6.2)	38.2 (8.6)	0.15
Marital status, number (%)			
Single	16 (59.3)	18 (78.3)	0.18
Married	8 (29.6)	2 (8.7)	
Divorced or widowed	3 (11.1)	3 (13)	
Duration of psychosis, years (SD)	3.0 (1.9)	2.0 (1.2)	0.03
Drug, number (%)			
Aripiprazole	16 (53.9)	12 (52.2)	0.83
Paliperidone	11 (40.7)	11 (47.8)	
PANSS, mean (SD)	72.48 (26.03)	65.57 (24.97)	0.35
N/L ratio, mean (SD)	2.2 (1.3)	1.5 (0.7)	0.03

**Table 2 brainsci-14-00602-t002:** Full ANCOVA for N/L ratio.

Variable	Sum of Squares	df	F	*p*-Value
Group	8.781	1	7.685	0.008
Age	0.187	1	0.164	0.688
Duration of psychosis	1.781	1	1.559	0.219
PANSS	1.607	1	1.406	0.243
Gender (male)	0.378	1	0.330	0.569
Marital status (married)	1.617	1	1.415	0.241
Marital status (single)	0.479	1	0.420	0.521
Drug (paliperidone)	0.036	1	0.032	0.860
Residual	46.852	41		

## Data Availability

The data presented in this study are available on request from the corresponding author. The data are not publicly available due to patients privacy.

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
