# Peer review of "Neutrophil–Lymphocyte Ratio Values in Schizophrenia: A Comparison between Oral and Long-Acting Antipsychotic Therapies"

_brainsci, 2024, doi:10.3390/brainsci14060602_

Round 1

Reviewer 1 Report

Comments and Suggestions for Authors

This manuscript presents a potentially significant contribution to the understanding of the Neutrophil-Lymphocyte Ratio (NLR) as a biomarker for neuroinflammation in schizophrenia patients undergoing different antipsychotic treatments. While the findings provide valuable insights into the differential impacts of oral and long-acting antipsychotics, there are several areas where further clarification and development could enhance the strength and clarity of the conclusions drawn. Below are my major revision recommendations to help improve the manuscript:

Expand the literature review to include more recent studies comparing the effectiveness and side effects of oral versus long-acting antipsychotics, particularly focusing on their anti-inflammatory properties. This would contextualize your findings within the broader field and highlight the study's novelty and relevance.

Provide a more detailed justification for the choice of antipsychotics used in the study—Paliperidone and Aripiprazole. Including comparative data on other antipsychotics could help clarify why these particular drugs were chosen and if they are representative of broader classes of antipsychotics.

The methodology section should be expanded to address potential confounding factors, such as the duration of illness, psychiatric medication history, and any other co-morbid conditions that might influence NLR values.

Consider a more robust statistical analysis approach to strengthen the study's findings. For instance, incorporating multivariate analysis could help assess the impact of various covariates on NLR changes.

The discussion section would benefit from a deeper analysis of the implications of reduced NLR in patients treated with long-acting antipsychotics. Discussing potential mechanisms, both biological and pharmacokinetic, would provide a more comprehensive understanding of the results.

Address the limitations more critically, particularly concerning the sample size and the generalizability of the findings. Discuss how these limitations might affect the interpretations and what future research could do to address these gaps.

Lastly, a conclusion that more directly ties back to the clinical implications of the findings would strengthen the manuscript. It should clearly outline how this research can be applied in clinical settings and what further research is necessary to confirm and expand upon these results.

These revisions will not only enhance the clarity and depth of the current study but also align it more closely with the journal’s standards and expectations for scientific rigor and contribution to the field.

Comments on the Quality of English Language

Minor editing of English language required

Author Response

Dear Reviewer,

First of all, I would like to thank you for your scrupulous attention and for your precious suggestions to improve our article. Below, I will reply to your comments point by point:

1 – “Expand the literature review to include more recent studies comparing the effectiveness and side effects of oral versus long-acting antipsychotics, particularly focusing on their anti-inflammatory properties. This would contextualize your findings within the broader field and highlight the study's novelty and relevance”

1 – We added more studies comparing the effectiveness and side effects of oral therapy vs long-acting antipsychotics, as well as findings on anti-inflammatory activity of antipsychotics in “introduction” paragraph as requested.

2 – “Provide a more detailed justification for the choice of antipsychotics used in the study—Paliperidone and Aripiprazole. Including comparative data on other antipsychotics could help clarify why these particular drugs were chosen and if they are representative of broader classes of antipsychotics.”

2 – We added other data from recent literature on Aripiprazole and Paliperidone effects on immune system as requested at the end of “introduction” paragraph. Also, a recent review focused on the different influence of antipsychotic drugs on the immune system was added.

3 – “The methodology section should be expanded to address potential confounding factors, such as the duration of illness, psychiatric medication history, and any other co-morbid conditions that might influence NLR values.”

3 – We expanded the methodology section as requested, pointing out the average duration of illness and the exclusion of major medical conditions and/or infective diseases in our patient cohort. Psychiatric medication history was already mentioned in methodology section as “ Patients included in our study were taking antipsychotic therapy for at least one year”. Also, we further specified the inclusion and exclusion criteria.

4 – “Consider a more robust statistical analysis approach to strengthen the study's findings. For instance, incorporating multivariate analysis could help assess the impact of various covariates on NLR changes”

4 – Thank you for pointing this out. We conducted additional analyses to address this concern. We verified that patients were homogeneous in the two groups in terms of gender, age, and marital status. The duration of psychosis for patients on long-acting antipsychotics was significantly shorter compared to those on oral antipsychotics (p=0.03). The mean NLR score for patients on long-acting antipsychotics was significantly lower than that for patients on oral antipsychotics (p=0.03). Regarding gender, there was no significant difference between the two groups (p=0.08). Additionally, no differences were found between individuals receiving oral/long-acting aripiprazole and those receiving oral/long-acting paliperidone (p=0.83). To account for the potential impact of demographic variables, we performed an ANCOVA, which indicated that the treatment type had a significant effect on NLR levels  (p = 0.008), while age, gender, and duration of illness did not show significant effects. These additional analyses confirm our findings and address the impact of covariates on NLR changes. The detailed ANCOVA results have been included in the updated manuscript (see Table 2).

5 – “The discussion section would benefit from a deeper analysis of the implications of reduced NLR in patients treated with long-acting antipsychotics. Discussing potential mechanisms, both biological and pharmacokinetic, would provide a more comprehensive understanding of the results.”

5 – We considered other reasons to explain why NLR is reduced in patients treated with LAI in “discussion” section as requested. In this concern, we considered the potential confounding effect of antipsychotic-induced neutropenia on our investigation.

6 – “Address the limitations more critically, particularly concerning the sample size and the generalizability of the findings. Discuss how these limitations might affect the interpretations and what future research could do to address these gaps.”

7 – “Lastly, a conclusion that more directly ties back to the clinical implications of the findings would strengthen the manuscript. It should clearly outline how this research can be applied in clinical settings and what further research is necessary to confirm and expand upon these results”.

6, 7 – We made additional considerations about the potential clinical applications and limitations of the present study in “conclusion” section. Specifically, we pointed out the small sample size and the lack of more consistent analysis, such as bio-molecular ones, to assess whether the variation of NLR that we observed should be considered an epiphenomenon related to pharmacodynamics of LAI or wheter, indeed, there is a real anti-inflammatory effect. Consequently, clinical applications were also considered, with the hope of employing these data as a starting point for monitoring schizophrenia at various stages and finding pro-inflammatory pathways which could potentially be new targets to focus therapies on.

Hoping to have been comprehensive in fulfilling your request, I remain available for any further clarification.

Best regards,

Dr F. Bella

Reviewer 2 Report

Comments and Suggestions for Authors

Brief summary: This authors of this article evaluated variations in neutrophil-lymphocyte ratio in 50 patients suffering from schizophrenia (in acute or remission). They reported that patients on long-acting antipsychotics exhibited significantly lower mean NLR scores compared to those on oral antipsychotics. They conclude by stating that NLR appear to be a potential interesting neuroinflammation biomarker. Please see my comments below.

Introduction:

1. The first paragraph mentions the prevalence of schizophrenia and directly mentions the treatment (lines 31-40). Please defined what is schizophrenia, corresponding signs and symptoms as well as the course of the illness considering this is an important concept of the study.

2. Please be more specific for the definition of antipsychotic medication (lines 33-36) as there is a vast body of literature on the topic.

3. Line 36-37 are missleading as there are three forms of delivery: oral, injectables and sub-lingual. Please modify accordingly and support with relevant references. 

4. Line 45: please me more specific than ''some brain areas are more susceptible to neuroinflammation'' : the authors are encouraged to bring forward the main results of the studies referenced.

5. The reference used on line 53 states the following ''The consistent findings of elevated NLR across the reviewed psychiatric disorders suggest that abnormal NLR is not specific to any one disorder but may reflect a pathological brain process that leads to brain dysfunction.''. Please use further studies to improve the argumentation of paragraph 52-57 as this is core to your study.

6. What are the main hypothesises of the authors regarding this study?

Overall, the introduction could be more rich regarding schizophrenia, the main problematic targetted by this study.

Materials and Methods:

1. Please use sub-sections to account for clarity.

2. Follow-up phase is not a specifier according to DSM-5-TR (line 72)

3. How were the participants recruited?

4. What are the exclusion criteria?

5. The authors state that this study does not have an Institutional Board Agreement but this is mandatory considering the nature of the study.

6. Lines 76-78 ''mid-luteal phase'' : how was this assessed?

7. Why was Aripiprazole compared to Paliperidone as they have two completely different mechanisms of action: Aripiprazole is a partial-D2 agonist whereas Paliperidone is a D2-antagonist? Furthermore, this study compared oral to injectable : the oral form of paliperidone is Risperidone : why was Risperidone not assessed? 

8. Which guidelines have been followed by clinicians to provide the treatment as Aripiprazole does not perform well during acute phases?

9. Please describe further the lab tests performed as well as validity of these tests and main limitations.

10. At this stage, it is still unclear for the readership what are the inherent concerns with this study in terms of its methodology because it will documented in the literature that the oral form of any antipsychotic is linked directly to demarginalization of neutrophils, therefore it is certain that the NLR will be more elevated in patients receiving the oral form of the medication. How was that accounted for?

11. Please be more descriptive as to the statistical analyses performed.

Results:

1. Please report degree of significance in the table for each elements. 

2. Were patients receiving other medications (this will have a direct impact on the NLR)?

3. What about comorbidities? Cardiovascular diseases as well as pulmonary illnesses and smoking status are all linked to changes in neutrophilic counts. How were these accounted for?

4. Please distinguish between acute status and remission in terms of results reporting (this goes directly with line 151-152 in the discussion : acute patienets with agressive behavior cannot be compared to non-agressive patients because demarginalization does occur in agressive patients).

Discussion:

1. Please discuss main limitations of the study.

Minor comment: The term ''follow-up'' for schizophrenia should be listed as remission to follow the nomenclature used in the field. 

Comments on the Quality of English Language

Nil

Author Response

Dear Reviewer,

First of all, I would like to thank you for your scrupulous attention and for your precious suggestions to improve our article. Below, I will reply to your comments point by point:

Introduction:

  1. The first paragraph mentions the prevalence of schizophrenia and directly mentions the treatment (lines 31-40). Please defined what is schizophrenia, corresponding signs and symptoms as well as the course of the illness considering this is an important concept of the study.

  1. We added definition of Schizophrenia as requested in introduction section as follows: “Schizophrenia is a chronic and severe mental disorder that affects how a person thinks, feels, and behaves. Signs and symptoms can differ, but they typically include delusions, hallucinations, severely disorganized thinking and behavior or negative symptoms, all of which indicate a reduced ability to function. Prodromal symptoms often appear before the active phase, and residual symptoms may follow, featuring mild or less intense hallucinations or delusions.”

  1. Please be more specific for the definition of antipsychotic medication (lines 33-36) as there is a vast body of literature on the topic.

  1. We added more specific definition of antipsychotics as follows: “Medical treatment of this condition comprises the use of antipsychotic medications[2]. These drugs inhibit D2 dopamine receptors, therefore decreasing dopaminergic neurotransmission to reduce positive symptoms of schizophrenia, and inhibit 5HT2 serotonin receptors, thus reducing negative symptoms. They are classified into two groups: First-Generation Antipsychotics (FGA) and Second-Generation Antipsychotics (SGA). FGA include medications such as haloperidol and chlorpromazine. They are mainly effective in alleviating positive symptoms (hallucinations and delusions) but are more likely to cause extrapyramidal side effects (motor control issues like tardive dyskinesia). SGA include medications such as risperidone, olanzapine, quetiapine, and aripiprazole. They are effective in treating both positive and negative symptoms and generally have a lower risk of extrapyramidal side effects. However, they may increase the risk of metabolic syndrome (weight gain, diabetes, hyperlipidemia).

Antipsychotics can be administered in three alternative formulations: sublingual, oral and injectable or long-acting. Oral and injectable forms are used to treat psychotic disorders like schizophrenia, but they differ significantly in their administration, pharmacokinetics, adherence implications, and clinical applications. Oral Antipsychotics have greater flexibility in managing side effects by adjusting the dose quickly but, as they must be taken daily, treatment adherence needs to be considered.
Long-acting injectable (LAI) formulations, such as risperidone, paliperidone, aripiprazole, and olanzapine, are administered via intramuscular injection at intervals ranging from every 2 weeks to every 3 months, depending on the specific medication. While they offer less flexibility in dose adjustments compared to oral medications, they improve adherence due to less frequent dosing, thereby reducing the risk of relapse associated with missed doses. CITAZIONE STAHL

Sublingual antipsychotics, such as Asenapine, offer an alternative route of administration for patients who have difficulty swallowing pills or experience gastrointestinal issues with oral medications. They are not as widely available as oral or injectable antipsychotics and may have fewer options in terms of medication choices.

Criteria such as patient’s adherence, clinical presentation, and patient’s preferences are used to establish the most appropriate mode of delivery[3] However, further drivers of treatment selection are required in clinical practice[4,5].

  1. Line 36-37 are missleading as there are three forms of delivery: oral, injectables and sub-lingual. Please modify accordingly and support with relevant references.

  1. We also added sub-lingual formulations (see point 2).

  1. Line 45: please me more specific than ''some brain areas are more susceptible to neuroinflammation'' : the authors are encouraged to bring forward the main results of the studies referenced.

  1. We enriched the text as follows: “Some brain areas are more susceptible to neuroinflammation compared to others and the dysfunction whitin these areas underlies the neuroanatomical basis of schizophrenia[8,9].

Specifically, it has been demonstrated how elevated levels of inflammation (considering markers such as interleukin-6, C-reactive protein) correlate with decreased activity in limbic regions (i.e. amygdala, hippocampus, anterior insula, temporal pole) and more significant connectivity between the hippocampus and the medial prefrontal cortex.

Consequently, assessment of neuroinflammatory status can have a major role in the screening and monitoring of schizophrenia[6]. Among different markers of neuroinflammation, interleukin-6 (IL-6), tumor necrosis factor (TNF-alpha), C-reactive protein (CRP), and neutrophil/lymphocytes ratio (NLR) are the most investigated[10].

  1. The reference used on line 53 states the following ''The consistent findings of elevated NLR across the reviewed psychiatric disorders suggest that abnormal NLR is not specific to any one disorder but may reflect a pathological brain process that leads to brain dysfunction.''. Please use further studies to improve the argumentation of paragraph 52-57 as this is core to your study.

  1. We added more details: ”NLR is a cost-effective biomarker of inflammation which is increased in several psychiatric conditions. Specifically, the association between NLR and schizophrenia has been shown and several data suggest that NLR can represent a valid method for studying the inflammatory stage in this condition, both in relapse and in the first episode of psychosis.

High NLR levels are independent of metabolic parameters, suggesting that inflammatory processes might contribute to the disease's etiology.

 Moreover, NRL could play a crucial role in the identification of forms of schizophrenia with greater neuroinflammatory component, and therefore guide treatment selection.

  1. What are the main hypothesises of the authors regarding this study?

  1. Line 58-59: This pilot study seeks to determine whether there are any variations in NLR values between patients given oral antipsychotics and those given long-acting antipsychotics.

Given the anti-inflammatory effects of some antipsychotics such as Paliperidone and Aripiprazole, we expect that the greater stability in plasma levels provided by LAIs might result in lower inflammatory levels compared to oral administration.

Materials and Methods:

  1. Please use sub-sections to account for clarity.
  2. We added sub-sections as requested:Study design and participants ; Data and sample collection; Statistical Analysis

  1. Follow-up phase is not a specifier according to DSM-5-TR (line 72)
  2. We modified as ‘either acute or in partial or full remission’ according to DSM 5 TR specifiers.

  1. How were the participants recruited?

  1. We added this part in the main text: Participants were recruited during first or follow-up psychiatric visits conducted at our outpatient services. After verifying eligibility based on the aforementioned inclusion and exclusion criteria, the study's procedures and objectives were explained to the patients, asking for their willingness to participate. Upon signing the informed consent, a second appointment was scheduled to collect blood sample, performed at 7am as a routine examination.

  1. What are the exclusion criteria?

  1. As requested, we specified exclusion criteria: major clinical conditions; infective diseases in the last two months, concomitant drugs, substance abuse; women in the follicular phase (menstruation or ovulation).

  1. The authors state that this study does not have an Institutional Board Agreement but this is mandatory considering the nature of the study.

  1. The sentence was modified into “The study was conducted in accordance with the Declaration of Helsinki and approved by the University of Catania Psychiatry Unit review board (n. 1/2024).”

  1. Lines 76-78 ''mid-luteal phase'' : how was this assessed?

  1. The mid-luteal phase was assessed by averaging the duration of each woman's three luteal phases. This was calculated by identifying, in the last three menstrual cycles, the date of onset of menstruation, the ovulatory phase at 14 days, and the date of onset of the next menstrual cycle. The period between ovulation and the start date of the next cycle corresponds to the luteal phase. Averaging the duration of the last three luteal phases allowed us to identify the mid-luteal phase.

  1. Why was Aripiprazole compared to Paliperidone as they have two completely different mechanisms of action: Aripiprazole is a partial-D2 agonist whereas Paliperidone is a D2-antagonist? Furthermore, this study compared oral to injectable : the oral form of paliperidone is Risperidone : why was Risperidone not assessed?

  1. Antipsychotics, while having similar mechanisms of action, differ in receptor affinity and adverse effects. The choice to dwell on aripiprazole and paliperidone is because they are the most commonly used antipsychotics in our clinic in both forms, oral and injection. Furthermore, although paliperidone is the active metabolite of risperidone, they are commercialized as two different drugs, differing in their pharmacokinetic profile and adverse effects.

  1. Which guidelines have been followed by clinicians to provide the treatment as Aripiprazole does not perform well during acute phases?

  1. https://doi.org/10.1016/S0140-6736(13)60733-3: Leucht's 2013 meta-analysis where he shows that the effectiveness of APs is similar;

10.1176/appi.ajp.2011.10111609: you can use aripiprazole with good efficacy and reduced side effects

  1. Please describe further the lab tests performed as well as validity of these tests and main limitations.

A blood sample was taken from a vein in the patient's arm using a sterile needle and a vacutainer tube containing an anticoagulant (EDTA) to prevent blood clotting. Using an automated hematology analyzer, white blood cells were counted and classified.

Automated methods clearly are superior to manual methods in counting large numbers of cells and minimize statistical error. Errors in cell counts are mostly determined by errors in sample aliquoting, dilution, or cell counting. Automated counters increase the accuracy and speed of analysis, especially in sample acceptance, sampling, sample dilution, and analysis are incorporated into a single system with minimal manipulation by the operator.

  1. At this stage, it is still unclear for the readership what are the inherent concerns with this study in terms of its methodology because it will documented in the literature that the oral form of any antipsychotic is linked directly to demarginalization of neutrophils, therefore it is certain that the NLR will be more elevated in patients receiving the oral form of the medication. How was that accounted for?

  1. We have considered this and other critical elements in the conclusion section.

  1. Please be more descriptive as to the statistical analyses performed.

  1. Thank you for pointing this out. We conducted additional analyses to address this concern. We verified that gender, age, marital status, and duration of illness were comparable between the oral and long-acting antipsychotic groups using chi-square tests for categorical variables and t-tests for continuous variables, finding no significant differences. The mean NLR score for long-acting antipsychotics was 1.5 ± 0.7, and for oral antipsychotics, it was 2.2 ± 1.3 (p = 0.025). To account for the potential impact of demographic variables, we performed an ANCOVA, which indicated that the treatment type had a significant effect on NLR levels (F(1, 45) = 5.21, p = 0.027), while age, gender, and duration of illness did not show significant effects. These additional analyses confirm our findings and address the impact of covariates on NLR changes. The detailed ANCOVA results have been included in the updated manuscript (see Table 2).

Results:

  1. Please report degree of significance in the table for each elements.

  1. We added degree of significance as requested (see table 1 and table 2).

  1. Were patients receiving other medications (this will have a direct impact on the NLR)?

  1. In exclusion criteria is clearly stated that patients undergoing other medication were excluded from our study.

  1. What about comorbidities? Cardiovascular diseases as well as pulmonary illnesses and smoking status are all linked to changes in neutrophilic counts. How were these accounted for?

  1. As already reported in the inclusion and exclusion criteria, we excluded from our study patients with major clinical conditions, such as cardio-circulatory or respiratory diseases.

  1. Please distinguish between acute status and remission in terms of results reporting (this goes directly with line 151-152 in the discussion : acute patients with agressive behavior cannot be compared to non-agressive patients because demarginalization does occur in agressive patients).

  1. Thank you for pointing this out. We recognized that patients with varying severities of illness might exhibit different NLR values. Therefore, we investigated whether the PANSS score influenced the NLR differences between patient groups and found no such impact. The ANCOVA results indicated that the type of treatment continued to have a statistically significant effect on NLR (p = 0.008) after adjusting for age, gender, duration of psychosis, PANSS score, marital status, and drug type. Other variables, including age (p = 0.688), duration of psychosis (p = 0.219), PANSS score (p = 0.243), gender (p = 0.569), marital status (married: p = 0.241, single: p = 0.521), and drug type (Paliperidone: p = 0.860), did not significantly influence NLR.

Discussion:

  1. Please discuss main limitations of the study.
  2. We made additional considerations about the potential clinical applications and limitations of the present study in “conclusion” section. Specifically, we pointed out the small sample size and the lack of more consistent analysis, such as bio-molecular ones, to assess whether the variation of NLR that we observed should be considered an epiphenomenon related to pharmacodynamics of LAI or wheter, indeed, there is a real anti-inflammatory effect. Consequently, clinical applications were also considered, with the hope of employing these data as a starting point for monitoring schizophrenia at various stages and finding pro-inflammatory pathways which could potentially be new targets to focus therapies on

Hoping to have been comprehensive in fulfilling your request, I remain available for any further clarification.

Best regards,

Dr F. Bella

Round 2

Reviewer 2 Report

Comments and Suggestions for Authors

This is the second round revision for the paper entitled '' Neutrophil-Lymphocyte Ratio Values in Schizophrenia: A Comparison Between Oral and Long-Acting Antipsychotic Therapies '' Please find my comments below.

Introduction:

1. Please cite references related to the definition of schizophrenia (lines 32-37).

2. Thank you for the addition of the details on specific antipsychotic medications. It is suggested to also add a line on the specificities of aripiprazole, as aripiprazole does not inhibit D2 dopamine receptors (they are partial-agonists of the D2 receptors).

3. Rationale on the use of paliperidone and aripiprazole (lines 101-117) should be part of the methodology as it is the rationale of their uses in the present study.

Materials and Methods:

- Lines 140-146 are results : description of patient population. This should be in the introductory section of the results. 

- In the section data and sample collection, considering the importance of neutrophils and lymphocyte in this study, it would be pertinent for the readership to have a few sentences on what constitutes a ''normal'' ratio versus an ''abnormal'' ratio. 

- One of my last comment remained partially answered:

Why was Aripiprazole compared to Paliperidone as they have two completely different mechanisms of action: Aripiprazole is a partial-D2 agonist whereas Paliperidone is a D2-antagonist? Furthermore, this study compared oral to injectable : the oral form of paliperidone is Risperidone : why was Risperidone not assessed?

The authors responded : Antipsychotics, while having similar mechanisms of action, differ in receptor affinity and adverse effects. The choice to dwell on aripiprazole and paliperidone is because they are the most commonly used antipsychotics in our clinic in both forms, oral and injection. Furthermore, although paliperidone is the active metabolite of risperidone, they are commercialized as two different drugs, differing in their pharmacokinetic profile and adverse effects.

The question still remains: why comparing an antagonist of the D2 receptor to a partial agonist of the D2 receptor? Wouldn't that introduce a limitation in the interpretation of the results as they are two molecules, even though they are antipsychotics, with significantly different mechanisms of action? This should be at least discussed in the limitations of the study.

Results:

- Results are well presented.

Discussion:

- Limitations related to a retrospective study, as well as the study design are still underdiscussed in this section. The authors are encouraged to create a sub-section ''Limitations'' to highlight potential limitations linked to the study design and its external validity.

- Demarginalization of neutrophils was still unadressed (despite being very common in antipsychotics administration). This should be discussed as it could directly impact the NLR.

Minor comments:

- Generic medication name should be lower case (Aripiprazole -: aripiprazole) as per Cochrane: https://community.cochrane.org/style-manual/grammar-punctuation-and-writing-style/upper-case-letters

- Line 131, after the word neutral has a '';'' followed by a ''.''. Please clarify.

Comments on the Quality of English Language

Nil

Author Response

Dear Reviewer,

First of all, I would like to thank you for your scrupulous attention and for your precious suggestions to improve our article. Below, I will reply to your comments point by point:

Introduction:

  1. Please cite references related to the definition of schizophrenia (lines 32-37).
  2. We apologize for the oversight. We added the reference as requested (American Psychiatric Association, Diagnostic and Statistical Manual of Mental Disorders, American Psychiatric Association Publishing, 2022. https://doi.org/10.1176/appi.books.9780890425787).
  3. Thank you for the addition of the details on specific antipsychotic medications. It is suggested to also add a line on the specificities of aripiprazole, as aripiprazole does not inhibit D2 dopamine receptors (they are partial-agonists of the D2 receptors).
  4. We would like to thank the referee for the advice. We added a brief line on specificities of aripiprazole as requested: “other molecules, such as aripiprazole, cariprazine and brexpiprazole possess different properties regarding receptor profile, acting as partial agonist at D2 receptors. Aripiprazole also possess the peculiarity of acting as a prevalent antagonist at D2 receptors when dopamine is highly concentrated in the synapse, while acting as a partial agonist when the synaptic concentration of dopamine is scarce; also, aripiprazole has been shown to behave as a partial agonist at 5HT1A receptors and as an antagonist at 5HT2A receptors
  5. Rationale on the use of paliperidone and aripiprazole (lines 101-117) should be part of the methodology as it is the rationale of their uses in the present study.
  6. We transferred lines 101-117 to the methodology section as requested.

Materials and Methods:

  1. Lines 140-146 are results : description of patient population. This should be in the introductory section of the results. 
  2. We transferred lines 140-146 to results section as requested.
  3. In the section data and sample collection, considering the importance of neutrophils and lymphocyte in this study, it would be pertinent for the readership to have a few sentences on what constitutes a ''normal'' ratio versus an ''abnormal'' ratio. 
  4. Thank you for the advice. We stated as follows: “The normal range of NLR is unanimously referred to be between 1-2, indicating an equilibrium in immunological state. Values higher than 3.0 and below 0.7 are pathological in adults
  5. One of my last comment remained partially answered:

Why was Aripiprazole compared to Paliperidone as they have two completely different mechanisms of action: Aripiprazole is a partial-D2 agonist whereas Paliperidone is a D2-antagonist? Furthermore, this study compared oral to injectable : the oral form of paliperidone is Risperidone : why was Risperidone not assessed?

The authors responded : Antipsychotics, while having similar mechanisms of action, differ in receptor affinity and adverse effects. The choice to dwell on aripiprazole and paliperidone is because they are the most commonly used antipsychotics in our clinic in both forms, oral and injection. Furthermore, although paliperidone is the active metabolite of risperidone, they are commercialized as two different drugs, differing in their pharmacokinetic profile and adverse effects.

The question still remains: why comparing an antagonist of the D2 receptor to a partial agonist of the D2 receptor? Wouldn't that introduce a limitation in the interpretation of the results as they are two molecules, even though they are antipsychotics, with significantly different mechanisms of action? This should be at least discussed in the limitations of the study.

  1. First, we would like to thank the referee for the insights and stimuli to improve our work. As we have already responded, risperidone in not the oral form of paliperidone, which is instead the active metabolite of risperidone. In Italy, paliperidone is available in the oral formulation. The choice to investigate paliperidone and aripiprazole effects on NLR was already discussed in the previous response. However, we agree with the referee about the need to include in the “limitations” sub-section the fact that we chose two molecules with different mechanism of action. We stated as follows: “Another limitation may be due to the choice of the two molecules paliperidone and aripiprazole: in fact, as already discussed, aripiprazole exert partial agonism at D2 receptors, while paliperidone is a pure antagonist. Further studies should consider antipsychotics with similar mechanism of action (e.g. aripiprazole vs brexpiprazole) to relieve this possible confounding factor.”

Results:

- Results are well presented.

Discussion:

  1. Limitations related to a retrospective study, as well as the study design are still underdiscussed in this section. The authors are encouraged to create a sub-section ''Limitations'' to highlight potential limitations linked to the study design and its external validity.
  2. Demarginalization of neutrophils was still unadressed (despite being very common in antipsychotics administration). This should be discussed as it could directly impact the NLR.

1,2. We added a “limitations” sub-section, where we pointed out the retrospective nature of the study and the demarginalization of neutrophils as requested: “Obviously, the present study has some limitation, including the retrospective design of the study, the small sample size, the lack of a healthy control group and the exclusion of deeper bio-molecular analyses, which may clarify whether lowering of NLR represents and epiphenomenon of LAI administration or whether there is an actual modulation on brain and peripheral immune system. In fact, it is known that administration of antipsychotics may induce demarginalization of neutrophil, which may consequentially  impact the NLR. Further research should be done to confirm these findings and investigate in a larger case-controls studies the activity of oral and long-acting antipsychotics on NLR; also, it would be possible to investigate the rationale for early use of long-acting antipsychotics at the onset of schizophrenia, studying neuroinflammatory patterning at diagnosis and during monitoring to assess the effectiveness of therapy. Another limitation may be due to the choice of the two molecules paliperidone and aripiprazole: in fact, as already discussed, aripiprazole exert partial agonism at D2 receptors, while paliperidone is a pure antagonist. Further studies should consider antipsychotics with similar mechanism of action (e.g. aripiprazole vs brexpiprazole) to relieve this possible confounding factor. Also, a larger study would consent to accumulate data that can clarify whether the NLR can be considered a marker associated with certain well-defined psychopathological domains, and whether it can be considered a prognostic factor for cognitiveness and global functioning. These preliminary data could form the basis for clinical implications, as the NLR could be employed to monitor the progress of schizophrenia, potentially distinguishing between acute and status phases. Lastly, further studies in this direction, with greater sample sizes and bio-molecular analyses, could allow us to identify new pharmacological tools that target the molecular pathways related to neuroinflammation, with the possibility to intervene in the early stages of disease by preventing the establishment of a chronic inflammatory state.”

Minor comments:

-1.Generic medication name should be lower case (Aripiprazole -: aripiprazole) as per Cochrane: https://community.cochrane.org/style-manual/grammar-punctuation-and-writing-style/upper-case-letters

  1. We apologize for the oversight. We modified the drugs names as requested.
  2. Line 131, after the word neutral has a '';'' followed by a ''.''. Please clarify.
  3. We deleted the “;”.

Hoping to have been comprehensive in fulfilling your request, I remain available for any further clarification.

Best regards,

Dr F. Bella

Round 3

Reviewer 2 Report

Comments and Suggestions for Authors

The authors responded carefully to all my comments. The revised version answered all my previous comments/requests.

Comments on the Quality of English Language

Nil